# Validity of a Self-Assessment Skin Tone Palette Compared to a Colorimeter for Characterizing Skin Color for Skin Cancer Research

Michelle K. Martin, Tanzida Zaman, Amanda M. Okello and Leslie K. Dennis *

Department of Epidemiology and Biostatistics, Mel and Enid Zuckerman College of Public Health, Tucson, AZ 85724, USA

* Correspondence: ldennis@arizona.edu

**Abstract:** Our goal is to determine whether our objective 9-point Self-Assessment Skin Tone Palette (SASTP) is correlated with a colorimeter's assessment of a melanin index, so that Hispanic and Black people can be included in skin cancer research where scales were developed for White populations. Subjects were asked to self-identify their skin tones using the SASTP. This study assessed the criterion validity of the SASTP by measuring a range of skin colors compared to a melanin index reported from a colorimeter for the upper-inner arm (non-sun-exposed skin color), and the outer forearm (sun-exposed). Among 188 non-artificial tanners, 50% were White, 30% were Hispanic or White-Hispanic, and 20% were other racial categories. Meanwhile, 70% were female (30% male) and 81% were age 18–29 (19% age 30+). The mean melanin of the upper-inner arm decreased with lighter skin color and stronger tendency to burn. The SASTP in comparison to melanin index values was correlated for both the upper-inner arm ($r = 0.81$, $p < 0.001$) and the outer forearm ($r = 0.77$, $p < 0.001$). The SASTP provides a 9-point scale that can be considered as an alternative, less expensive method that is comparable to the objective colorimeter melanin index, which may be useful in studies on skin cancer among White, non-White, and Hispanic peoples.

**Keywords:** colorimetry; color palette; skin neoplasms; skin pigmentation

## 1. Introduction

Skin cancer is an important public health issue [1]. The main risk factors are ultraviolet radiation and skin sensitivity to the sun. Meta-analyses examining phenotypic factors or sun sensitivity and skin cancer (melanoma and basal cell skin cancer) indicated heterogeneity between reported relative risk estimates of the various studies [2,3]. This lack of consistent magnitude of odds ratios for risk of skin cancer and sun sensitivity may be attributed to differences in the definition or interpretation of sun sensitivity. Other potential reasons for differences include age, sex, and other demographics among the different populations pooled worldwide. Additionally, studies did not report the same categories, thus making pooling across different nominal scales difficult. Major differences were seen when comparing the magnitude of the odds ratios for sun sensitivity factors and melanoma between Hispanic [4] and White people [2] especially for fair, medium, or dark skin color. This may be real or due to differing interpretations of such a subjective scale by people of different racial and/or ethnic groups.

Skin cancer studies for White populations typically use self-assessment of sun sensitivity including 4-point scales of tendency to burn, 4-point scales of tanning ability, or combined Fitzpatrick self-assessed skin type to quantify sun sensitivity. Dermatologists often use the expanded V and VI category Fitzpatrick skin-type to include people of color [5]. A self-assessment of tendency to burn and tan was developed for White populations, and is often used in skin cancer studies restricted to White people. There is a moderate chance of misclassifying self-assessed exposures due to different interpretations of the wording of

sun sensitivity measures between participants or by the same participant at a later time [6]. Several reliability studies among White populations show a wide range of Kappa statistics ranging from 0.51–0.72 for tendency to burn, 0.59–0.76 for tanning ability, 0.53–0.79 for skin-type, and 0.69–0.78 for skin color [6–10]. Kappas are often labeled as very good or excellent above 0.8 and substantial or good above 0.6 to 0.8, fair or moderate for 0.4 to 0.6, but poor below 0.2 [11–14], so these results represented a fair to good reliability among White populations. Misclassification of sun sensitivity in questionnaires is largely due to misinterpretation of ambiguously worded descriptions of skin color and inconsistent wording between questionnaires [3,6,15]. Varying interpretations of wording and inconsistent reliability results for self-assessing risk indicate the need for a more concrete method of describing skin color, such as a more diverse color palette that could be used to better self-assess skin characteristics and skin cancer risk. Other skin color scales have been developed ranging from use of photography, 36 skin tones, or expanded scales only focused on Caucasian/White people.

The purpose of our study was to first expand traditional skin color measurements that only focus on non-Hispanic White people to a skin palette to include Hispanic and Black people, and to then assess its validity. To do so, we first assessed face validity of the initial and revised palette to develop our Self-Assessment Skin Tone Palette (SASTP). Secondly, for our main aim here, we used criterion validity, sometimes also referred to as inter-method reliability [16,17], to compare the SASTP to the melanin index obtained from a colorimeter across individuals of various racial/ethnic backgrounds.

## 2. Materials and Methods

### 2.1. The Self-Assessment Skin Tone Palette (SASTP) Development

2.1.1. Drafted Palette, Face Validity including Cognitive Interviewing Revisions

The SASTP was designed to incorporate skin tones with more variation than the Fitzpatrick skin type while being compact and easy for the participant to use. Skin tones were initially chosen similarly to the various skin tone options used for products in the make-up industry for various ethnicities. Then, such tones were created in Adobe Photoshop by varying the color to match. The process began with 24-point skin tones. Face validity, as a subset of content validity, was assessed by skin cancer experts examining the shades and number of skin tones present [16]. Next, cognitive interviewing of volunteers (staff/students/faculty at the College of Public Health born in the US, Mexico, and Africa) with different skin tones was conducted by asking if the skin palette shades were distinguishable enough from each other and fit the participants' skin tones [18]. This included volunteers originating from Nigeria and other parts of Africa, who suggested the darkest skin tone needed to be lighter. Similarly, other volunteers who considered themselves to be Hispanic/Latinx provided similar input for their skin tones, as did fair subjects who cannot tan. From this, a 9-point scale was determined because the color options were discriminant while allowing for the swatches to be easily viewable on a single sheet of paper.

2.1.2. SASTP Scale

The scale was set up in a $3 \times 3$ pattern with rows down A, B, C and columns across 1, 2, 3, where A1 was the lightest color and C3 was the darkest (Figure 1). Participants had the ability to hold up the palette to compare to their forearm and inner arm during data collection. The final SASTP 9 skin tones were translated to melanin values of 19.4, 20.5, 25.0, 30.6, 37.6, 58.6, 72.0, 90.5, and 102.2.

The outer forearm typically represents one of the more tanned areas on a person's body, whereas the area of the upper-inner arm receives less sun exposure [19]. Thus, we chose the forearm to represent sun exposed skin color and the upper-inner arm to represent unexposed skin color. These sites were used for both SASTP and colorimeter assessment.

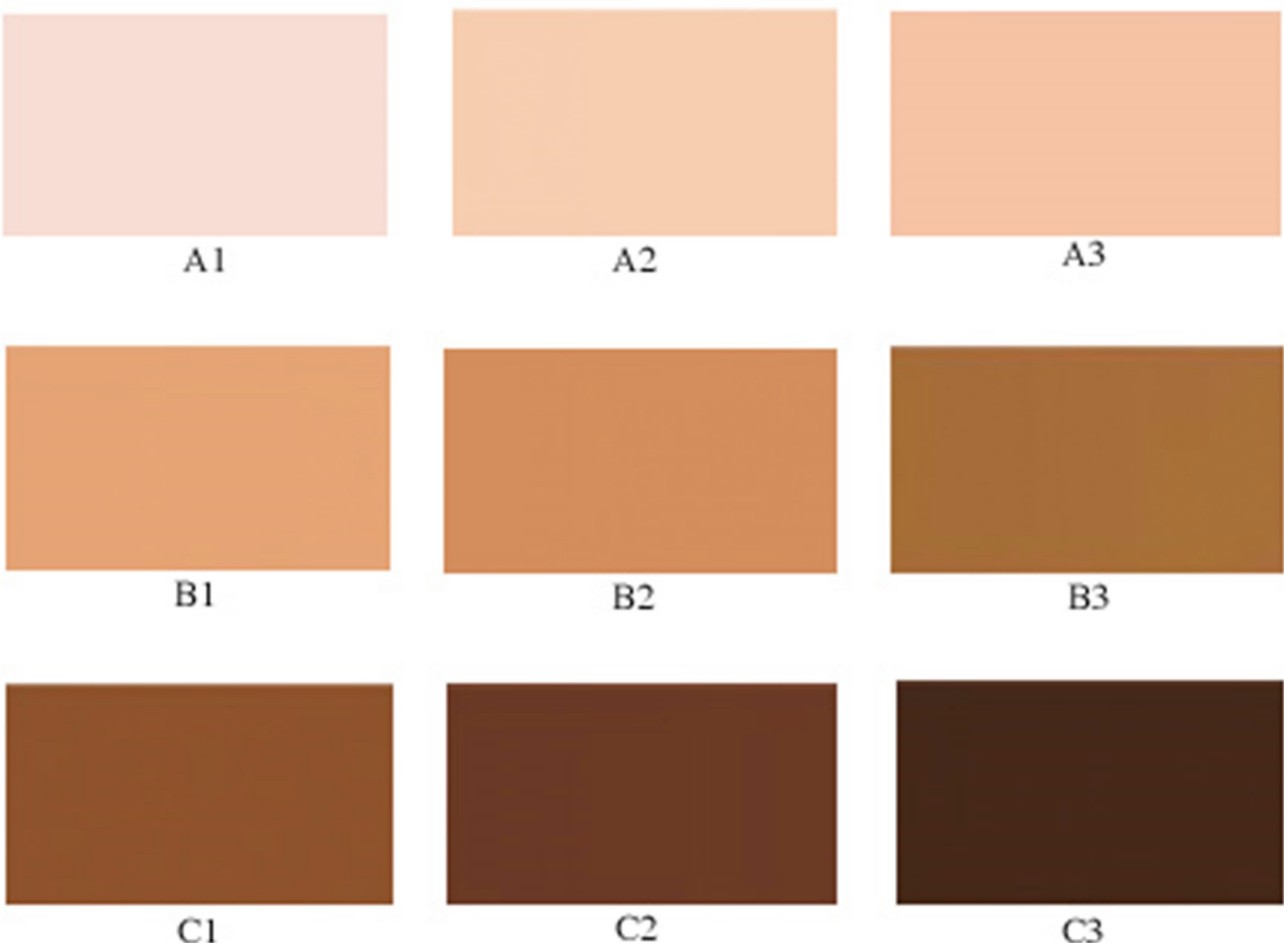

**Figure 1.** This presents the Self-Assessment of Skin Tone Palette (SASTP) with 9 skin tone shades from lightest A1, A2, A3, B1 . . . to C3.

*2.2. Colorimetry*

Colorimeters can measure melanin index and erythema index using spectrophotometry. The investigators used the DSM II ColorMeter (Cortex Technology, Hadsund, Denmark) to take objective measurements of the melanin index of participants' upper-inner arm and outer forearm. The melanin index measures the tan level of the skin. The erythema index measures the redness of the skin, but this can be affected by short-term changes in room temperature and blushing along with longer effects of a sunburn. Our focus was skin color, so the erythema index was not pertinent. The DSM II ColorMeter is a handheld colorimeter that is a reliable measure of melanin (M) index that uses tristimulus colorimetry and narrow-band spectrophotometry technology [20,21]. The colorimeter's measurement of a melanin index is discriminative and sensitive when measuring normal skin colors for White people and Black people with day-to-day repeatability [20]. Spectrophotometers or colorimeters are handheld devices that are not affected by ambient lighting [22]. To verify that the DSM II was not affected by ambient lighting, we used the colorimeter to conduct an experiment measuring the colors in the color palette at the same locations with the same lighting as our main study. We added an outdoor location and found the measurements within this reliability study of the colorimeter did not vary by location ($p = 0.99$) [23]. Studies have evaluated the reproducibility of related techniques of narrowband reflectance spectrophotometry and tristimulus colorimetry in field conditions [24,25]. Uter et al. [25] reported on reproducibility of field conditions of similar technology (a chromameter and a reflectometer) for a very large population.

*2.3. Survey*

A short questionnaire consisting of 15 questions was administered. The three questions about artificial tanning included using sunlamps, tanning beds, or sunbeds in the last year, in the last month, and average minutes of use. Those using a tanning bed in the last month were excluded from analyses as such use could actually change the melanin index, which would reduce the true comparison of measures. Sun sensitivity measures were also surveyed, including tendency to sunburn (on initial exposure to the sun in spring or summer), inability to tan (after repeated and prolonged exposure), skin color of the upper-inner arm (fair, medium or dark), skin color of the forearm in comparison with the upper inner arm, eye color, natural hair color at age 20, and family history of skin cancer (yes, no, do not know). The survey also asked about age (21–29, 30–39, 40–49, 50–59, 60+), sex, and race/ethnicity background. Racial/ethnic background was asked as one question, and respondents were instructed to check all that apply (Hispanic, Native American or Alaska native, African American/Black, Asian or Pacific Islander, or White), but most only checked one category. Thus, a White racial/ethnic background reflects non-Hispanic White, as most who checked Hispanic did not check a racial category. Skin color in two body sites based on the SASTP was included in the survey along with four questions to create a unique identification. The self-perception of skin tone was ascertained for two body sites with a color palette in which participants chose from nine skin tone choices which they believed to match their (1) upper-inner arm and (2) forearm. The survey and measurement took about 5 min.

*2.4. Recruitment*

For recruitment, we contacted staff, and faculty via email and/or flyers. Additionally, we setup a data collection booth at the annual Melanoma Walk to create awareness of melanoma on campus and recruited students within an undergraduate statistics course, collecting their data either before or after class on several occasions. The survey was administered from October–December in 2016 in Tucson, Arizona. We consented participants and collected data. The participants were allowed to withdraw at any time or refuse to answer any questions. For subjects who participated in the test–retest reliability, they were recontacted 3–5 weeks after their first in-person survey; some preferred to be recontacted via email and an online second survey. Data were entered and stored on password-protected computers, and physical copies of the questionnaires were kept locked in an investigator's office. The consent form and study procedures were approved by the Institutional Review Board for human research.

*2.5. Analysis*

Analyses excluded participants reporting use of tanning beds or artificial tanning sprays and creams in the prior month, as the artificial tan can influence the melanin index measurements. These excluded subjects were more likely to be age 18–29 and female. Mean values of melanin index obtained from the colorimeter were stratified by various self-reported measures of sun sensitivity, including the new skin tone palette.

2.5.1. Repeat Reliability

Repeat reliability was examined using an interclass correlation coefficient (ICC) [16]. The square root of the reliability coefficient is an estimate of the upper limit of validity based on the parallel test assumption [16]. Standard "rule of thumb" for interpreting correlations of >0.9 as very high or excellent, 0.8 to 0.9 as high or good, 0.7 to 0.8 as high or fair, 0.5–0.7 as moderate, 0.3–0.5 as low, and <0.3 as negligible correlations were used as descriptive adjectives [26–28].

As a scale cannot be valid if it is not reliable, thus we first examined its repeat reliability [16]. The SASTP had moderate to high reliability among 146 subjects who repeated questions about the color of their upper-inner arm and color of their forearm using the SASTP re-assessed one month later [18]. Half of these subjects completed the re-assessment

in-person and half completed it online. As measurements used the same scale, the best measure of repeated reliability is an ICC [16]. The test–retest reliability for the upper inner arm had ICCs ranging from 0.57–0.83, whereas the ICCs ranged from 0.71–0.72 for the forearm [18].

2.5.2. Criterion Validity/Intermethod Reliability

Criterion validity, also known as intermethod reliability [16,17,29], was ascertained by comparing the melanin index values reported from the colorimeter and the self-selected color from the 9-point color palette obtained at the same time. Intermethod reliability or validity is a measure of the ability of two different instruments measuring the same underlying concept to yield the same results on the same subjects. Validity is when one is comparing a more precise but more burdensome measurement to a simple measurement more easily used in a larger study [16]. Various statistical measures can be used in intermethod reliability/validity, dependent on the type of exposure measurement from continuous, dichotomous, nominal, or ordered category by, respectively, using Pearson correlations, sensitivity and specificity, misclassification matrices, or Spearman correlations [16].

In our study, the colorimeter is the more precise and more burdensome measure which produces values on a continuous scale. The 9-point SASTP scale is between an ordered categorical and continuous scale, which was developed based on skin tones translated via the colorimeter to a melanin scale. The Spearman correlation would compare the SASTP based on the rank order of the 9-values and the melanin values. Because the SASTP was not developed as a 9-point equally spaced scale or a rank ordered scale, the Pearson product-moment correlation is more appropriate for capturing the 9-points of the SASTP based on the melanin scale of the skin tones that they were created to match in comparison to the continuous melanin scale [16]. The SASTP 9-skin tones were translated to melanin values of 19.4, 20.5, 25.0, 30.6, 37.6, 58.6, 72.0, 90.5, and 102.2 based on colorimeter measurements of the skin tones chosen for the scale. SAS versions 9.4 and University (SAS Institute, Inc., Gary, NC, USA) were used for conducting data analysis to obtain correlations and 95 percent confidence intervals.

**3. Results**

Of the 188 participants, about 50% self-classified as White, 30% as Hispanic or White-Hispanic, 12% as Asian, 3% Native American or Alaskan, and 4% African American. Due to the size and demographics of the undergraduate class ($N = 144$), the majority of participants were 18–29 years of age (81%), with 5% ages 30–39, 1% ages 40–49, 8% ages 50–59, and 5% age 60 or older. Majority of our study participants were female (70%). Moreover, subjects excluded from validity analyses for use of tanning sprays or creams in the prior month were more likely to be younger, female, and more likely to burn. On average, female participants had slightly lower average melanin ($M$) index value ($M = 39.4$) than male participants ($M = 41.2$) for the upper-inner arm (unexposed), and even more so for average melanin of the forearm (exposed), with 46.2 for females compared to 52.7 for males.

Mean values and ranges of melanin index of the upper-inner arm and the forearm are presented in Table 1 by sun sensitivity factors to show how standard sun sensitivity measures relate to melanin. Both "tendency to burn" and skin color of the upper-inner arm are measures related to untanned skin color, so they are only compared to upper-inner arm melanoma. Whereas self-assessed forearm color compared to the inner arm is worded to be related to how much darker the sun-exposed forearm is, therefore, it is only compared to the melanin of the forearm. For tendency to sunburn after initial exposure to the sun, the average melanin index values gradually increased across the response categories as expected from 34.8 for subjects reporting a severe and painful sunburn to 42.2 for those who answered no burn. Mean melanin index values of the forearm decreased from 50.2 for an ability to tan deeply to 46.4 for tanning mildly; having no tan had the highest average melanin index value of 51.5. The gradient of melanin of the upper-inner arm did correspond with self-reported skin color of the upper-inner arm (Table 1). Average

melanin index values increased with darker eye color and hair pigmentation but were less clearly associated. White participants had the greatest variability of melanin index values with a range of 20 to 60. For the most part, melanin values were lightest in subjects who self-reported their background as White, followed by Hispanic-White, Hispanic, Asian, Native American and then African American or Black peoples. While our face validity and cognitive interviewing processes for developing the SASTP included subjects of a wide range of skin types, the survey and colorimeter data did not capture very many subjects who self-classified as Black or had melanin values above 70 (Table 1; Figure 2).

**Table 1.** Mean Colorimeter Values of Melanin and Erythema by Categories of Self-Reported Measures of Sun Sensitivity Among 188 Non-Artificial Tanners.

| Self-Assessed Risk Factors | Upper Inner Arm | | Forearm | |
|---|---|---|---|---|
| | N | Melanin (M): Mean (Range) | N | Melanin (M): Mean (Range) |
| Tendency to burn [1] | | | | |
|   Severe & painful sunburn | 8 | 34.79 | (29.12, 39.92) | | |
|   Moderate sunburn | 39 | 38.15 | (31.02, 89.75) | | |
|   Mild sunburn | 64 | 38.92 | (31.27, 76.30) | | |
|   No sunburn, just tans | 77 | 42.22 | (19.93, 90.72) | | |
| Inability to tan [2] | | | | |
|   Have no tan | 10 | 43.99 | (29.12, 89.75) | 10 | 51.51 | (33.26, 102.13) |
|   Mildly tanned | 47 | 39.34 | (31.27, 90.72) | 47 | 46.38 | (33.51, 90.72) |
|   Moderately tanned | 91 | 39.38 | (19.93, 76.30) | 91 | 47.85 | (36.88, 76.92) |
|   Deeply tanned | 40 | 40.88 | (33.15, 73.73) | 40 | 50.25 | (38.61, 85.16) |
| Skin Color of the upper inner arm (untanned) [3] | | | | |
|   Fair | 109 | 37.70 | (19.93, 89.75) | | |
|   Medium | 76 | 42.07 | (33.47, 66.21) | | |
|   Dark | 3 | 67.07 | (36.76, 90.72) | | |
| Forearm compared to upper inner arm | | | | |
|   About the Same | | | | 68 | 48.23 | (33.72, 102.13) |
|   A little darker | | | | 76 | 47.01 | (33.26, 64.14) |
|   Somewhat darker | | | | 41 | 49.94 | (40.41, 90.72) |
|   A lot darker | | | | 3 | 53.16 | (50.14, 55.37) |
| Eye Color | | | | |
|   Gray | 2 | 35.31 | (33.29, 37.33) | 2 | 43.60 | (37.17, 50.03) |
|   Blue | 30 | 35.43 | (29.12, 43.00) | 30 | 43.94 | (33.51, 60.00) |
|   Green | 25 | 35.19 | (19.93, 44.67) | 25 | 42.83 | (33.26, 58.12) |
|   Brown or Black | 128 | 42.10 | (32.11, 90.72) | 128 | 50.29 | (34.97, 102.13) |
| Natural Hair Color at age 20 | | | | |
|   Red | 2 | 33.78 | (31.08, 36.47) | 2 | 39.90 | (33.26, 46.53) |
|   Red-Blond | 4 | 32.56 | (29.12, 35.64) | 4 | 40.00 | (34.06, 46.26) |
|   Blonde | 18 | 36.38 | (31.92, 43.00) | 18 | 45.83 | (36.88, 54.83) |
|   Brown, or Auburn | 120 | 38.37 | (19.93, 89.75) | 120 | 46.31 | (33.51, 102.13) |
|   Black | 44 | 46.60 | (31.02, 90.72) | 44 | 55.39 | (40.47, 90.72) |
| Racial/Ethnic Background | | | | |
|   White | 95 | 36.10 | (19.93, 51.49) | 95 | 44.83 | (33.26, 60.00) |
|   Hispanic-White | 14 | 37.18 | (33.47, 43.00) | 14 | 43.77 | (36.26, 53.22) |
|   Hispanic | 42 | 41.93 | (34.63, 59.31) | 42 | 49.99 | (40.47, 64.14) |
|   Asian | 23 | 41.87 | (32.34, 53.13) | 23 | 50.23 | (39.89, 69.46) |
|   Native American | 6 | 43.61 | (39.78, 47.17) | 6 | 51.97 | (48.21, 59.23) |
|   Black | 8 | 71.53 | (53.10, 90.72) | 8 | 77.62 | (64.14, 102.13) |
| SASTP choice for the | | | | |
|   1 (A1) | 20 | 35.49 | (30.68, 39.50) | 1 | 38.43 | (38.43, 38.43) |
|   2 (A2) | 84 | 36.80 | (19.93, 51.49) | 33 | 41.61 | (33.51, 48.88) |
|   3 (A3) | 35 | 39.27 | (32.34, 45.55) | 41 | 44.63 | (33.26, 54.15) |
|   4 (B1) | 36 | 43.95 | (33.15, 66.21) | 47 | 45.69 | (36.88, 56.81) |
|   5 (B2) | 7 | 43.38 | (35.98, 53.13) | 53 | 52.85 | (39.89, 68.13) |

**Table 1.** *Cont.*

| Self-Assessed Risk Factors | Upper Inner Arm | | | Forearm | | |
|---|---|---|---|---|---|---|
| | N | Melanin (M): Mean (Range) | | N | Melanin (M): Mean (Range) | |
| 6 (B3) | 3 | 63.28 | (49.99, 73.73) | 8 | 55.64 | (42.31, 69.17) |
| 7 (C1) | 1 | 90.72 | | 3 | 81.78 | (69.46, 90.72) |
| 8 (C2) | 1 | 76.30 | | 1 | 76.92 | |
| 9 (C3) | 1 | 89.75 | | 1 | 102.1 | |

Abbreviations: N, sample size; SASTP, Self-Assessment of Skin Tone Palette [1] When your skin is first exposed to strong sunlight for an hour for the first time each spring or summer with no protection does it get ... [2] After repeated and prolonged exposure to the sun, does your skin become ... [3] How would you describe your un-tanned skin color on your upper inner arm?

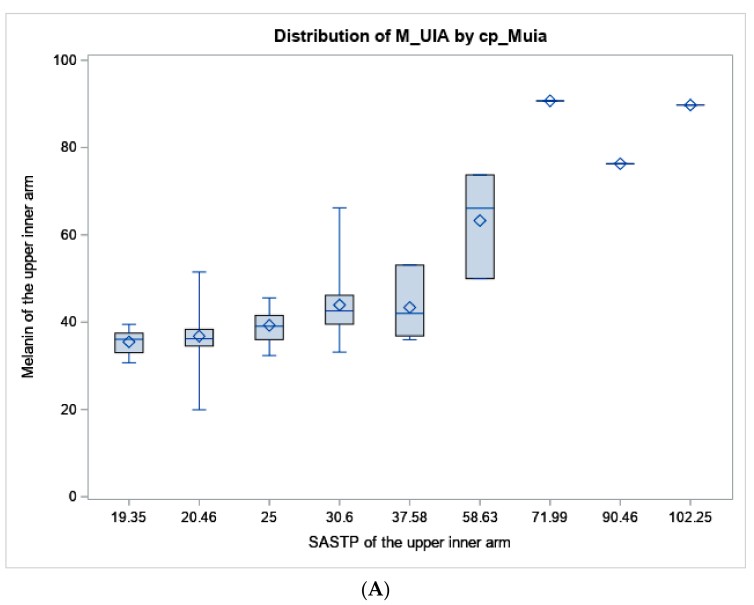

(**A**)

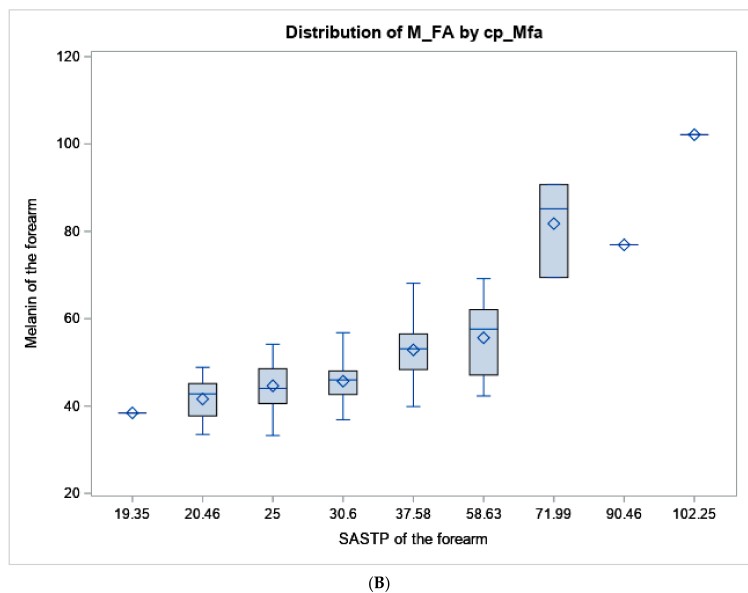

(**B**)

**Figure 2.** Boxplot of melanin values from the colorimeter vs. the SASTP for the (**A**) upper inner arm, and (**B**) forearm. The diamond represents the mean, the horizontal line in the box represents the median. the length of the box represents the interquartile range, the whiskers at the end of each represent the maximum and minimum values of melanin.

Criterion validity of the respondents' choices within the SASTP in comparison to melanin index values are reported in Table 2. Our newly developed self-assessment skin color palette values correlated well with colorimeter melanin index values for participants. This was true when comparing values from the palette and the colorimeter for the upper-inner arm and for the forearm with the exception of one subject who chose the 8th darkest color out of 9 colors, but this should be interpreted cautiously with only 1.6% of subjects choosing 7, 8, or 9.

**Table 2.** Repeat reliability and Criterion Validity (Intermethod Reliability) of the Self-Assessment of the Skin Tone Palette (SASTP) initially taken in-person among non-artificial tanners.

| Reliability/Validity Measure: Repeat Reliability of SASTP [1] | N | ICC | Lower Limit of ICC | Upper Limit of Validity [2] |
|---|---|---|---|---|
| Repeat survey taken online [3] | | | | |
| Upper inner arm (unexposed skin color) | 72 | 0.83 | 0.76 | 0.91 |
| Forearm (exposed skin color) | 72 | 0.71 | 0.60 | 0.83 |
| Repeat Survey taken in-person [3] | | | | |
| Upper inner arm (unexposed skin color) | 73 | 0.57 | 0.43 | 0.77 |
| Forearm (exposed skin color) | 73 | 0.72 | 0.62 | 0.87 |
| **Skin Tone Palette vs. Colorimeter Melanin Intermethod Reliability/Criterion Validity [4]** | **N** | **Corr** | **95% CI** | |
| Upper inner arm (unexposed skin color) | 188 | 0.81 | 0.75–0.85 | |
| Forearm (exposed skin color) | 188 | 0.77 | 0.70–0.82 | |

Abbreviations: CI, confidence interval; Corr, Pearson product-moment correlation; ICC, interclass correlation coefficient; N, number of subjects; SASTP, Self-Assessment of Skin Tone Palette; [1] Instructions: Using the color palette provided, what color best matches the skin color of your . . . ; [2] The upper-limit of validity was estimated based on the reliability; [3] Test–retest reliability of the Self-Assessment of Skin Tone Palette (SASTP) based on an Interclass correlation coefficient (ICC); ICC reliability used to estimate upper limit of validity; [4] Intermethod Reliability/Criterion Validity comparing the SASTP values translated to melanin values to the melanin values from a colorimeter measurement using the Pearson Product-Moment correlation.

## 4. Discussion

Our SASTP was developed to include Hispanic people and others in skin cancer research that is often restricted to non-Hispanic White people. The SASTP was designed to have skin tones that sought to match the melanin in one's skin color, particularly among White and Hispanic people. We found that the SASTP is an accurate tool for self-assessment of skin tone compared to an objective measure of tanness of the skin (the colorimeter) whether looking at untanned skin (upper-inner arm) or tanned skin (forearm). Of our non-artificial tanners, 50% self-classified other than non-Hispanic White people. We saw overlapping melanin values for skin color among those who self-classified their background as Hispanic, Hispanic-White, or White.

Robinson et al. [30] used a culturally adaptive Fitzpatrick skin-type classification based on descriptions by Eilers et al. [31] among adults attending Latino community health fairs in Chicago. They found significant differences in the melanin values for the adapted Fitzpatrick skin-types II, III verses IV, and IV verses V, using a spectrophotometric assessment. Similar to our findings of overlapping melanin values, they showed overlapping categories of skin-type for Latinos who were Mexican–American (skin-type: 3% I, 42% II, 44% III, 10% IV) and Puerto Rican (skin-type: 4% II, 48% III, 44% IV) [30]. Our study saw similar distributions of melanin among Hispanic people.

The development of the SASTP was predicated to help skin cancer studies be more inclusive. Melanoma studies in particular have tended to focus on only White populations based on much higher rates of melanoma among White people (27.2/100,000 compared to 4.8/100,000 among Hispanic people) [32]. Limited studies of skin cancer have been conducted among Hispanic people [33]. Despite this lower incidence rates of melanoma in Hispanic–American people, they are 2.4 times more likely to present at a later stage, and 3.6 times more likely to develop distant metastases [34–37]. While some assume that the lower incidence of melanoma among Hispanic people is due to protection from darker

skin, studies have shown it may also be due in part to behavioral differences including lower indoor tanning use and lower intentional tanning than among non-Hispanic White people [38,39]. However, many sun sensitivity factors related to melanoma in White populations have also been shown to be important among Hispanic populations including fair skin color, skin type, and sunburns [4]. Skin cancer surveillance behaviors occur among few Hispanic people [33], which is likely related to later stage at diagnosis seen among Hispanic populations. Furthermore, many Hispanic people do not perceive that they are at risk of skin cancer despite knowing skin cancer can occur in any skin tone [40]. Changing erroneous beliefs that Hispanic or Latinx people are at no risk of melanoma is important for risk awareness and early detection of melanoma [30].

In skin cancer research, sun sensitivity measured via tendency to sunburn, inability to tan, and skin color (fair, medium or dark) each signify about a 2-fold increase in risk [2,3]. When asked about the depth of their tan after repeated and prolonged exposure, the melanin index values for subjects who reported to have "no tan" ($M = 44$) were higher than those "deeply tanned" ($M = 41$). This reflects what has been seen in other studies that the "no tan" category likely includes both people who do not tan (because they are very fair) and people with naturally dark skin that do not attempt to tan, making a category of "no tan" non-predictive of skin cancer due to heterogeneity. For more accurate risk assessment, the diverse groups within "no tan" need to be separated, but often are not.

Early studies of skin tones used von Luschan tiles, a set of 36 tiles created in the 1950s, to match skin tones [41]. A study of 246 volunteers aged 18–40 from Pennsylvania and Atlanta, Georgia had the tiles matched to four body locations by two observers and had their melanin scored for each body location using a spectrophotometer. The Pearson's correlation coefficient comparing the validity of the skin color predicted by the tiles with measured melanin was 0.95 with *p*-value < 0.00001 representing differences less than 1 tile [41]. A second study in Thailand, thus including a narrower range of skin tones, found a correlation of 0.90 among 52 participants [42]. However, the use of 36 skin tone categories can be laborious. A Skin Color Chart® computed from skin reflectance spectrum was developed by L'Oreal using the Chromasphere® based on parameters of the CIELAB 1976 System, but requires their machinery [43]. Others used the Munsell system of soil color charts that was revised to consist of color-chips arranged in a three-dimensional expression of concepts of hue, value, and chroma [44,45]. Skin reflectance was measured via spectrophotometer, and reflectance was converted to an Individual Typology Angle (ITA) score which is a measure of reflectance from pure white to black [44]. The ITA value is an objective skin color type. A validity study of Munsell Soil Color Charts reported validity with spectrophotometer that calculated ITA values for an ICC of agreement of 0.61 [44]. A study of skin color of schoolchildren used 40 of the possible 141 Munsell color tiles and, compared to self-reported skin color, found that the mean ITA was different for white, light brown, and brown skin [45]. However, using these tiles is still laborious and requires an interviewer.

The Norwegian Women and Cancer study examined test–retest reliability of their 10-point skin color scale, reporting reliability of 0.59 (95% CI of 0.55–0.63) [46]. Unfortunately, they did not provide much information on their scale. Our scale had higher test–retest reliability. The research interviewer in an Australian study classified schoolchildren by natural skin, hair, eye color, and ethnicity, and used two charts for skin color assessment [47]. Chart 1 was based on the 5-point Fitzpatrick skin-type scale. Chart 2 had 10 skin-types. When both charts were modeled together, the Fitzpatrick skin-type scale accounted for 39.7% of variability, whereas the 10-point scale for Chart 2 explained 21.4% [47]. A study on college-aged participants in the Philadelphia area in spring of 2007. Students answered standard sun sensitivity questions and scored their untanned skin color on a 7-point order-ranked scale, along with having an observer rate their skin color on a 6-point scale and having their skin color from light to dark measured using spectrophotometers report of L* where 0 is pure black and 100 is pure white [22]. They found the observer's measure inversely correlated with the L* (since L* goes from black to white) for the lower arm and the check.

Similarly, self-rating of natural skin color was inversely correlated ($r = -0.70$) [22]. However, it was unclear at what body site natural skin color was assessed by the individual and the observer, whereas the article clearly stated that the spectrophotometer measured skin color on the lower arm and the cheek. Our self-reported skin tone in comparison to melanin index values had criterion validity not quite as high as the Lucian tiles, but stronger than that reported for the Munsell system. Furthermore, our SASTP did not require trained research staff to evaluate skin color with 36–40 tiles.

Strengths of our study include a color palette with a wide range of skin tones while still being easily accessible and a quick measurement tool, a sample size > 175 subjects, and an objective measure of skin color (colorimeter) to validate the SASTP. The population included a reasonable number of subjects that self-identified as White and Hispanic, along with a few other subjects who self-classified as Asian, Native American, or African American. The shade-inclusive color palette had a wide variety of shades that were easily distinguishable from one another.

The study population was predominately White and/or Hispanic (80%). Thus, more research to investigate self-assessment of SASTP in darker skinned individuals would expand the usefulness of our 9-point skin tone palette. Only three participants selected skin tones C1-C3 that were in the "dark" skin tone category. Having a more diverse study population with more participants for each of the nine skin tones would provide a more accurate representation of the full-scale. However, most populations do not evenly distribute on any scale. Other limitations of this study include a younger population, therefore, additional research among older subjects is warranted. Although measurements using the colorimeter were taken in varying types of natural and artificial light, manufacturers of the DSM II ColorMeter claim that the brightness of bulb compensates for differences in lighting during data collection [48]. To verify this, we used the colorimeter to test the colors on the color palette in the same locations with the same lighting and added on outdoor location and found the measurements did not vary by location ($p = 0.99$) [23].

## 5. Conclusions

Our SASTP was shown to be an alternative, less expensive method that is comparable to a colorimeter within our predominantly White and Hispanic population. It may be useful to explore different options for displaying the skin tone palette such as skin tone options that are not in rows and columns of increasing shades, but are linear (all in one row). This could make it easier for participants to choose the color closest to their skin color. Directions for further research would include validating the darker end of the skin tone scale and ordering the palette for skin tone linearly. More information on the validity of self-assessment measures for risk factors of skin cancer is needed for prevention and health promotion.

**Author Contributions:** Conceptualization, M.K.M. and L.K.D.; methodology, M.K.M., T.Z., A.M.O. and L.K.D.; formal analysis, M.K.M., T.Z. and A.M.O.; data curation, M.K.M., T.Z., A.M.O. and L.K.D.; writing—original draft preparation, M.K.M., T.Z. and A.M.O.; writing—review and editing, M.K.M., T.Z., A.M.O. and L.K.D.; project administration, M.K.M., T.Z., A.M.O. and L.K.D. All authors have read and agreed to the published version of the manuscript.

**Funding:** This research received no external funding.

**Institutional Review Board Statement:** This research was reviewed and approved by the University of Arizona's Institutional Review Board and determined to be exempt. The study was conducted in accordance with the Declaration of Helsinki and approved by the Institutional Review Board.

**Informed Consent Statement:** Subjects were provided elements of consent for these anonymously collected.

**Data Availability Statement:** These data were collected with IRB approval as part of course so were not publicly.

**Conflicts of Interest:** The authors declare no conflict of interest.

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
