# Peer review of "Validity of a Self-Assessment Skin Tone Palette Compared to a Colorimeter for Characterizing Skin Color for Skin Cancer Research"

_curroncol, doi:10.3390/curroncol30030241_

Round 1
Reviewer 1 Report
The topic of this paper is interesting and important but perhaps would be more appropriate for a dermatology journal. The methods and writing are weak.
Intro
1. This sentence is unclear: “Meta-analyses examining phenotypic factors or sun sensitivity and skin cancer (melanoma and basal cell skin cancer) indicated heterogeneity between reported odds ratios of the various studies [2,3].” I think you are referring to associations with skin cancer risk/incidence, not odds ratios per se.
2. “This lack of consistent magnitude of odds ratios for risk of skin cancer and sun sensitivity may be attributed to differences in the definition or interpretation of sun sensitivity.” There are other possible reasons than those mentioned, e.g., demographics of the sample, seasonality, etc.
3. “Several reliability studies among Whites show a wide range of Kappa statistics, that are used to compare questions from a test-retest reliability study for categorical data rather than the correlations used to compare continuous data, including from 0.51-0.72 for tendency to burn, 0.59-0.76 for tanning ability, 0.53-0.79 for skin-type and 0.69-0.78 for 47 skin color [6-10].” The focus of the sentence should be on the strength of the associations rather than the statistic reported. Same for comment #1.
4. Face validity (subset of content validity) and criterion validity are not “used” they are assessed, evaluated, etc. Very little info about the procedures for assessing face validity was provided.
Methods
1. More should be said about how the scale was developed, i.e., rationale for RGB, about Photoshop, makeup options. Also the CI and adjustments that were made.
2. To me, the palette looks somewhat limited. It seems very pinkish and doesn’t seem to include olive-toned skin.
3. The test-retest reliability should be a separate part of the methods and results. If measures are not reliable, they cannot be valid.
4. The colorimetry should be a separate part of the Methods. How does the melanin index relate to RGB? Does the melanin index measure natural skin color, tan-ness, and/or sunburns? Were measurements only taken once? I assume “collecting their data before and after class on several occasions” refers to different rather than the same students.
5. More detail should be provided about the survey items in the Methods including referring to Table 1.
6. The purpose of the study doesn’t seem to be entirely consistent with the analyses. I recommend adding a measurement expert to the team.
7. Why was melanin stratified rather than using continuous data? Not sure why kappas were used since that is typically used for different raters of the same thing not different measures of similar things. Hypotheses should be put with the purpose of the study in the Intro.
Results - The description of the participants in the Methods should be switched with the one in the Results.
Table 1 – No rationale was given for why some measures were included for one part of the body and not others. It looks like “Background” refers to race and ethnicity. Clarify that white means WNH and that Hispanic means HNW. When the study refers to Hispanics, it’s not always clear whether this refers to NWH or white Hispanics or both. There are so few people who chose some of the palette choices that I’m not sure they should be included in the same way as the others, and stronger limitations on this topic should be discussed.
Table 2 is unnecessary.
Discussion
1. This is a non-sequitur and needs further discussion: “Changing erroneous beliefs that Hispanics or Latino’s [sic] are at no risk of melanoma are [sic] important for risk awareness and early detection of melanoma [26].”
2. Some of the prior studies could have been briefly mentioned in the Intro to justify the contribution of the current study. There is no definition or context for LCH. The Munsell section needs work.
3. How would this be used beyond U of A? Different printers would print the colors differently. Can this be used digitally?
Other
1. One paper you may want to take a look at is Comparing alternative methods of measuring skin color and damage. Daniel LC, Heckman CJ, Kloss JD, Manne SL. Cancer Causes Control. 2009 Apr;20(3):313-21. doi: 10.1007/s10552-008-9245-3. Epub 2008 Oct 18. I suspect there are a number of others that would also be relevant.
2. Proofread for typos, etc. throughout.
Author Response
MDPI reviewer #1
Comments and Suggestions for Authors
The topic of this paper is interesting and important but perhaps would be more appropriate for a dermatology journal. The methods and writing are weak.
Intro
- This sentence is unclear: “Meta-analyses examining phenotypic factors or sun sensitivity and skin cancer (melanoma and basal cell skin cancer) indicated heterogeneity between reported odds ratios of the various studies [2,3].” I think you are referring to associations with skin cancer risk/incidence, not odds ratios per se.
Per the reviewer’s comments, we have changed this to “relative risk estimates”.
- See the revised manuscript with tracked changes version, line 30.
- “This lack of consistent magnitude of odds ratios for risk of skin cancer and sun sensitivity may be attributed to differences in the definition or interpretation of sun sensitivity.” There are other possible reasons than those mentioned, e.g., demographics of the sample, seasonality, etc.
Per the reviewer’s comment, we have added:
“Other potential reasons for differences such as age, sex and other demographics among the different populations pooled worldwide. Additionally, studies did not report the same categories making pooling across different nominal scales difficult.”
- See the revised manuscript with tracked changes version, lines 32-35.
- “Several reliability studies among Whites show a wide range of Kappa statistics, that are used to compare questions from a test-retest reliability study for categorical data rather than the correlations used to compare continuous data, including from 0.51-0.72 for tendency to burn, 0.59-0.76 for tanning ability, 0.53-0.79 for skin-type and 0.69-0.78 for 47 skin color [6-10].” The focus of the sentence should be on the strength of the associations rather than the statistic reported. Same for comment #1.
We have changed this to:
“Several reliability studies among Whites populations show a wide range of Kappa statistics, that are used to compare questions from a test-retest reliability study for categorical data rather than the correlations used to compare continuous data, including ranging from 0.51-0.72 for tendency to burn, 0.59-0.76 for tanning ability, 0.53-0.79 for skin-type and 0.69-0.78 for 47 skin color [6-10].”
- See the revised manuscript with tracked changes version, lines 48-50.
- Face validity (subset of content validity) and criterion validity are not “used” they are assessed, evaluated, etc. Very little info about the procedures for assessing face validity was provided.
Per the reviewer’s comments, we changed “used” to “assessed” but then clarified the purpose of the study to say:
“The purpose of our study was to first expand traditional skin color measurements, only focused on non-Hispanic Whites, to a skin palette to include Hispanics and Blacks, to then assess its validity. To do so, we first assessed face validity of initial and revised palettes to develop our Self-Assessment Skin Tone Palette (SASTP). Secondly, for our main aim here, we used criterion validity, also sometimes referred to as intermethod reliability [16-17], to compare the SASTP to the melanin index obtained from a colorimeter across individuals of various racial/ethnic backgrounds.”
- See the revised manuscript with tracked changes version, lines 62-70.
Methods
- More should be said about how the scale was developed, i.e., rationale for RGB, about Photoshop, makeup options. Also the CI and adjustments that were made.
Per the reviewer’s suggestion, we expanded the Methods regarding the SASTP to how it was drafted and adjustments that were made for the final scale. Then we examined reliability via correlation coefficients and 95% confidence intervals.
- See the revised manuscript with tracked changes version, lines 73-102, & 226-8.
- To me, the palette looks somewhat limited. It seems very pinkish and doesn’t seem to include olive-toned skin.
The palette was developed from make-up skin tones for fair, medium/olive, and dark skin tones. It was then reviewed by people of varying skin tones for further recommendations. This has been added to the methods.
- The test-retest reliability should be a separate part of the methods and results. If measures are not reliable, they cannot be valid.
We agree with the reviewer that if the methods are not reliable, they cannot be valid. Thus, we published the reliability study first (elsewhere) but briefly describe and reference it here.
- See the revised manuscript with tracked changes version, lines 183-196.
- The colorimetry should be a separate part of the Methods. How does the melanin index relate to RGB? Does the melanin index measure natural skin color, tan-ness, and/or sunburns? Were measurements only taken once? I assume “collecting their data before and after class on several occasions” refers to different rather than the same students.
Per the reviewer’s suggestion, we separated the colorimetry within the Methods section and explained the melanin index of the tanness or brownness of the skin.
We clarified that we collected the data from students either before or after class.
- See the revised manuscript with tracked changes version, lines 115-135, 168.
- More detail should be provided about the survey items in the Methods including referring to Table 1.
Per the reviewer’s suggestion, we have added more detail about the survey items in the Methods.
- See the revised manuscript with tracked changes version, lines 136-163.
- The purpose of the study doesn’t seem to be entirely consistent with the analyses. I recommend adding a measurement expert to the team.
We appreciate the reviewers comment and have rewritten the purpose to properly reflect the goals of this manuscript.
- See the revised manuscript with tracked changes version, lines 62-70.
We developed this tool as part of an exposure measurement course. We re-reviewed the manuscript and the methods from the book and made necessary revisions. Additionally, we re-verified our methods with epidemiological measurement experts and a biostatistician.
- Why was melanin stratified rather than using continuous data? Not sure why kappas were used since that is typically used for different raters of the same thing not different measures of similar things. Hypotheses should be put with the purpose of the study in the Intro.
Melanin is a continuous factor and treated as such, thus means are provided in Table 1. These means are stratified by various self-reported measures sun sensitivity factors to be able to view how melanin varies by such factors.
- See the revised manuscript with tracked changes version, lines 246-251.
Kappas were not used here, thus the erroneous sentence from our previous reliability study was removed. We only mention Kappas from other publications.
Additionally, we have rewritten the purpose to properly reflect the goals of this manuscript (lines 62-70).
Results - The description of the participants in the Methods should be switched with the one in the Results.
Per this and other reviewers’ comments, we reduced the redundancy.
Table 1 – No rationale was given for why some measures were included for one part of the body and not others. It looks like “Background” refers to race and ethnicity. Clarify that white means WNH and that Hispanic means HNW. When the study refers to Hispanics, it’s not always clear whether this refers to NWH or white Hispanics or both. There are so few people who chose some of the palette choices that I’m not sure they should be included in the same way as the others, and stronger limitations on this topic should be discussed.
Per the reviewer’s comment, we have added a rational for when comparisons are included: “Both tendency to burn and skin color of the upper inner arm are measures related to untanned skin color so are only compared to upper inner arm melanoma. Whereas self-assessed forearm color compared to the inner arm is worded to be related to how much darker the sun-exposed forearm is, therefore, it is only compared to the melanin if the forearm.”
We have clarified that the question about one’s background measures is a combined race/ethnicity question that asked participants to check all that apply.
- We clarified this in lines 147-151.
We agree that while we developed the SASTP for all skin types including African, the participants in Arizona did not include very many African Americans/Blacks; furthermore, some who identified as African American/Black did not perceive their skin color in the darker categories. Thus, we stated that the upper categories of the SASTP should be interpreted with caution and recommend further research. But due to overlap in categories there was no scientific way to exclude parts of the scale.
Table 2 is unnecessary.
We agree that the text duplicates the information in Table 2, but as these are our main findings, we think it is important to include Table 2. Thus, we have reduced the text and modified the title of Table 2 to correspond with what had been stated in the text.
Additionally, we added the repeat reliability correlations to Table 2.
Discussion
- This is a non-sequitur and needs further discussion: “Changing erroneous beliefs that Hispanics or Latino’s [sic] are at no risk of melanoma are [sic] important for risk awareness and early detection of melanoma [26].”
We greatly appreciate the reviewer for pointing this out. We have expanded this part of the discussion to talk about the lower rates of melanoma, UVR behavioral differences and knowledge of skin cancer risk among Hispanics that led to the sentence.
- See the revised manuscript with tracked changes version, lines 315-331.
- Some of the prior studies could have been briefly mentioned in the Intro to justify the contribution of the current study. There is no definition or context for LCH. The Munsell section needs work.
We very briefly mentioned other scales in the Introduction and significantly expanded the discussion of other scales in the Discussion.
- See the revised manuscript with tracked changes version, lines 351-358, 363-366, & 370-380.
Based on the reviewer’s comments we have removed the mention of L, C , H from the original discussion point as it is not central to this manuscript and distracts from the point of describing other skin color charts. We have expanded and clarified the discussion on Munsell Charts (lines 351-358).
However, in adding the Philadelphia study that reports L* we define it in the text (lines 375-76).
- How would this be used beyond U of A? Different printers would print the colors differently. Can this be used digitally?
The SASTP was developed based on national and international colors of skin (lines 80-88). This can be used digitally.
Other
- One paper you may want to take a look at is Comparing alternative methods of measuring skin color and damage. Daniel LC, Heckman CJ, Kloss JD, Manne SL. Cancer Causes Control. 2009 Apr;20(3):313-21. doi: 10.1007/s10552-008-9245-3. Epub 2008 Oct 18. I suspect there are a number of others that would also be relevant.
Thank you. We have added the Daniels manuscript along with 2 others (lines 126-135) but have focused the manuscript of the validity of the SASTP based on the colorimeter.
- Proofread for typos, etc. throughout.
Thank you, we have done so.
Reviewer 2 Report
Dear authors,
Your article brought an interesting idea on this subject, but your article needs further editing.
- The material method part should be more detailed,
- Discussion part is short and shallow
Best wishes...
Author Response
MDPI reviewer #2
Comments and Suggestions for Authors
Dear authors,
Your article brought an interesting idea on this subject, but your article needs further editing.
- The material method part should be more detailed,
We have expanded the methods and added sub-headings to help clarify detail.
-See lines 73-88 on developing the SASTP, 115-135 on colorimetry, lines 136-151 on the survey, and lines 201-220 on criterion validity.
- Discussion part is short and shallow
We expanded the discussion regarding development of the SASTP (lines 315-329), to better describe the Munsell charts (lines 349-356), and added other studies (lines 363-6, 370-70).
Reviewer 3 Report
In epidemiological studies of melanoma and keratinocyte carcinoma, determination of skin type plays a very important role. Self-assessment of skin type via skin tone reported in self-administered questionnaires always has a subjective component as long as clear guidance is lacking on how to classify specific skin tones. This manuscript addresses this problem with particular emphasis on the fact that especially Hispanics have problems of adequate mapping in classifications of skin type originally developed for Whites. The authors have developed the SASTP instrument, a color-mapped way of determining skin type in self-administered questionnaires, which they present in this manuscript and whose intermethod reliability they examine in a study of moderate size. They present the results of this study in the manuscript and discuss the implications of their results for future epidemiologic studies in this area.
Specific remarks:
To describe the agreement between SASTP and the melanin index measured by the DSM II ColorMeter the authors use the Pearson correlation coefficient. If I understood it correctly, the SASTP is measured on an ordinal 9-point scale. How are the nine ordinal categories transformed into a metric scale, i.e. what numeric values have been assigned to the nine categories? In addition, the authors should give a rationale why they measure agreement between an ordinal variable and a metric variable by the Pearson correlation coefficient that is suited to capture the strength of a linear association between two metric variables. To my opinion, other measures like the Spearman correlation coefficient or Kendall's tau would be better suited here.
What is the rationale for giving p-values in addition to point estimates of the Pearson correlation coefficient in the manuscript at several places (in the abstract, Table 2)? A statistical test of independence does not make any sense in a situation where the association between two variables aiming to measure the same thing has to be evaluated. Did you really expect that these variables were independent from each other? The low p-values only indicate that the SASTP is not independent from the melanin index, a fact which is beyond doubt due to the definition of the variables. A much more informative way of presenting the results would have been the addition of confidence intervals to show the precision of point estimates.
Correlation coefficients are not proportions, therefore in line 176 the percentage signs are misleading.
Why does Table 1 give no information on mean colorimeter values for the forearm in subcategories of the variable tendency to burn?
A nice addition to Table 1 would be a figure showing box plots or (better) violin plots for the distribution of colorimeter values in the subgroups defined by SASTP categories. Table 1 gives only the mean and the range, the figure would provide more detailed information.
The information on the type of participants in line 136/7 of the Results section is a repetition of the information in the section 2.4 of the Methods section. Add numbers for the different subtypes of participants (or omit the redundant information).
When summarizing results from previous studies on the reliabilty of different variables trying to capture skin type in epidemiologic studies, the authors missed the study by Veierod et al. (Reproducibility of self-reported melanoma risk factors in a large cohort study of Norwegian women. Melanoma Res. 2008 Feb;18(1):1-9. doi: 10.1097/CMR.0b013e3282f120d2) which should be added.
A more detailed description of the validity of the DSM II ColorMeter in measuring the melanin index is required in section 2.3 since this is the reference method in the evaluation of intermethod reliability. The authors cite only two validation experiments and omit more critical evaluations pointing to substantial measurement variability of such devices (e.g. Uter et al. Assessing skin pigmentation in epidemiological studies: the reliability of measurements under different conditions. Skin Res Technol. 2013 May;19(2):100-6. doi: 10.1111/srt.12013.). A recent systematic review on a related topic might also be of interest for the authors (Langeveld M at al. Skin measurement devices to assess skin quality: A systematic review on reliability and validity. Skin Res Technol. 2022 Mar;28(2):212-224. doi: 10.1111/srt.13113.). The Discussion has to reflect on these aspects as the choice of the reference method has a direct impact on the results of the study on intermethod reliability.
Finally, a thorough proofreading of the short manuscript needs to be done. There are several sentences that are not complete in the current form (probably one word is missing) and therefore hard to understand. Plural and singular are interchanged, abbreviations are introduced more than once, sample sizes are refered to with capital N and lower case n, etc. etc.
Overall, I think the topic of the paper is of high relevance and the study evaluating SASTP is also interesting. Although results from an evaluation of the SASTP have been published in previous papers by the group, the manuscript adds the aspect of intermethod reliability to the evaluation. However, the manuscript has quite a few deficiencies in its presentation outlined above, making it impossible to accept the manuscript for publication in this form.
Author Response
MDPI reviewer #3
Comments and Suggestions for Authors
In epidemiological studies of melanoma and keratinocyte carcinoma, determination of skin type plays a very important role. Self-assessment of skin type via skin tone reported in self-administered questionnaires always has a subjective component as long as clear guidance is lacking on how to classify specific skin tones. This manuscript addresses this problem with particular emphasis on the fact that especially Hispanics have problems of adequate mapping in classifications of skin type originally developed for Whites. The authors have developed the SASTP instrument, a color-mapped way of determining skin type in self-administered questionnaires, which they present in this manuscript and whose intermethod reliability they examine in a study of moderate size. They present the results of this study in the manuscript and discuss the implications of their results for future epidemiologic studies in this area.
Specific remarks:
To describe the agreement between SASTP and the melanin index measured by the DSM II ColorMeter the authors use the Pearson correlation coefficient. If I understood it correctly, the SASTP is measured on an ordinal 9-point scale. How are the nine ordinal categories transformed into a metric scale, i.e. what numeric values have been assigned to the nine categories? In addition, the authors should give a rationale why they measure agreement between an ordinal variable and a metric variable by the Pearson correlation coefficient that is suited to capture the strength of a linear association between two metric variables. To my opinion, other measures like the Spearman correlation coefficient or Kendall's tau would be better suited here.
We appreciate the reviewer’s comments. For Intermethod reliability (thus using different scales), the Pearson correlation coefficient is for continuous data and the Spearman rank correlation for ordered categories. Our data have both. Additionally, the 9-point ordered category in not based on rank order or equally spaced categories but based on melanin values from the initial created skin tones. The Kendall’s Tau is also a nonparametric measure that ignores the spacing between the chosen melanin values for the SASTP by only ranking the data. All 3 of these measures are measures of Intermethod reliability (White et al., Principles of exposure measurement in epidemiology: Collecting, evaluating, and improving measures of disease risk factors. 2nd ed. Oxford: Oxford University Press; 2008). However, we believe data is lost by rank ordering just as it would be lost comparing the colorimeter melanin data to a scale from 1-9. Thus, we added more detail about the melanin values upon which the SASTP was based (9 skin tones were translated to melanin values of 19.4, 20.5, 25.0, 30.6, 37.6, 58.6, 72.0, 90.5 and 102.2), and the Pearson correlation analyzes this.
- See the revised manuscript with tracked changes version, lines 198-228.
What is the rationale for giving p-values in addition to point estimates of the Pearson correlation coefficient in the manuscript at several places (in the abstract, Table 2)? A statistical test of independence does not make any sense in a situation where the association between two variables aiming to measure the same thing has to be evaluated. Did you really expect that these variables were independent from each other? The low p-values only indicate that the SASTP is not independent from the melanin index, a fact which is beyond doubt due to the definition of the variables. A much more informative way of presenting the results would have been the addition of confidence intervals to show the precision of point estimates.
P-values for correlations are not statistical tests of independence. P-values for correlations are testing the correlation not being due to random chance. However, we are now reporting confidence intervals for the Intermethod reliability in Table 2.
Correlation coefficients are not proportions, therefore in line 176 the percentage signs are misleading.
Thank you for pointing out this typo. It is appreciated. We changed this to 0.77 and 0.81, but later removed it as it is redundant of Table 2.
Why does Table 1 give no information on mean colorimeter values for the forearm in subcategories of the variable tendency to burn?
Per the reviewer’s comment, we have added a rational for when comparisons are included: “Both tendency to burn and skin color of the upper inner arm are measures related to untanned skin color so are only compared to upper inner arm melanoma. whereas self-assessed forearm color compared to the inner arm is worded to be related to how much darker the sun-exposed forearm is so is only compared to the melanin if the forearm.”
- See the revised manuscript with tracked changes version, lines 246-251.
A nice addition to Table 1 would be a figure showing box plots or (better) violin plots for the distribution of colorimeter values in the subgroups defined by SASTP categories. Table 1 gives only the mean and the range, the figure would provide more detailed information.
We have added a box plot for melanin of the upper inner arm and one for the forearm as Figure 2.
The information on the type of participants in line 136/7 of the Results section is a repetition of the information in the section 2.4 of the Methods section. Add numbers for the different subtypes of participants (or omit the redundant information).
We appreciate the reviewer’s suggestion and have removed the redundancy.
When summarizing results from previous studies on the reliabilty of different variables trying to capture skin type in epidemiologic studies, the authors missed the study by Veierod et al. (Reproducibility of self-reported melanoma risk factors in a large cohort study of Norwegian women. Melanoma Res. 2008 Feb;18(1):1-9. doi: 10.1097/CMR.0b013e3282f120d2) which should be added.
Thank you for this, we have added it and at least one other article. This article is focused of test-retest reliability of sun sensitivity factors including freckling and nevi along with a 10-point skin color scale that is not described. We have added it to the discussion of test-retest reliability of skin color scales, but it does not present or reference its validity.
- See the revised manuscript with tracked changes version, lines 363-366, and 370-380.
A more detailed description of the validity of the DSM II ColorMeter in measuring the melanin index is required in section 2.3 since this is the reference method in the evaluation of intermethod reliability. The authors cite only two validation experiments and omit more critical evaluations pointing to substantial measurement variability of such devices (e.g. Uter et al. Assessing skin pigmentation in epidemiological studies: the reliability of measurements under different conditions. Skin Res Technol. 2013 May;19(2):100-6. doi: 10.1111/srt.12013.). A recent systematic review on a related topic might also be of interest for the authors (Langeveld M at al. Skin measurement devices to assess skin quality: A systematic review on reliability and validity. Skin Res Technol. 2022 Mar;28(2):212-224. doi: 10.1111/srt.13113.). The Discussion has to reflect on these aspects as the choice of the reference method has a direct impact on the results of the study on intermethod reliability.
We have expanded the detail under the DSM II ColorMeter which included studies specifically of the DSM II. We conducted our own experimental study of the DSM II under different conditions that replicated the condition within our main study and add an outdoor location and found the measurements within this reliability study of the colorimeter did not vary by location (p=0.99). These studies are referenced when specifically talking about the DSM II.
- See the revised manuscript with tracked changes version, lines 115-135.
Thank you for the 2022 systematic review which includes the study by Uter et al., (2013). Langeveld et al., (2022) reported on several mexameters, camera-based colorimeters and similar devices evaluated in predominantly White populations. Uter et al., (2013) reported on reproducibility in field conditions of a chromameter and a reflectometer, the DSM II is related but slightly different technology. We have mentioned these but focus on the evaluations of the DSM II including our experiment using the DSM II with a wider range of skin tones.
- See the revised manuscript with tracked changes version, lines 115-135.
Finally, a thorough proofreading of the short manuscript needs to be done. There are several sentences that are not complete in the current form (probably one word is missing) and therefore hard to understand. Plural and singular are interchanged, abbreviations are introduced more than once, sample sizes are refered to with capital N and lower case n, etc. etc.
Thank you for your comments. We corrected the lower case “n” for sample size, spell-checked, fixed plurals and proofread the revised manuscript.
Overall, I think the topic of the paper is of high relevance and the study evaluating SASTP is also interesting. Although results from an evaluation of the SASTP have been published in previous papers by the group, the manuscript adds the aspect of intermethod reliability to the evaluation. However, the manuscript has quite a few deficiencies in its presentation outlined above, making it impossible to accept the manuscript for publication in this form.
Thank you for supporting this research. The prior publications were specifically looking at the 1) reliability of the DSM II colorimeter in different conditions, and 2) the reliability of the SASTP as it cannot be valid if it is not reliable. This has been clarified in the methods section.